# *Candida albicans* Hexokinase 2 Challenges the *Saccharomyces cerevisiae* Moonlight Protein Model

**DOI:** 10.3390/microorganisms9040848

**Published:** 2021-04-15

**Authors:** Romain Laurian, Jade Ravent, Karine Dementhon, Marc Lemaire, Alexandre Soulard, Pascale Cotton

**Affiliations:** 1INSA Lyon, CNRS, Université de Lyon, Université Claude Bernard Lyon1, UMR5240 MAP, 69622 Villeurbanne, France; romain.laurian@gmail.com (R.L.); jade.ravent@univ-lyon1.fr (J.R.); marc.lemaire.bio@univ-lyon1.fr (M.L.); alexandre.soulard@univ-lyon1.fr (A.S.); 2UMR-CNRS 5234, Laboratoire de Microbiologie Fondamentale et Pathogénicité, Université de Bordeaux, 33076 Bordeaux, France; karine.dementhon@u-bordeaux.fr

**Keywords:** *Candida albicans*, *Saccharomyces cerevisiae*, hexokinase 2, glucose repression, hexose kinase activity, hyphal transition, virulence

## Abstract

Survival of the pathogenic yeast *Candida albicans* depends upon assimilation of fermentable and non-fermentable carbon sources detected in host microenvironments. Among the various carbon sources encountered in a human body, glucose is the primary source of energy. Its effective detection, metabolism and prioritization via glucose repression are primordial for the metabolic adaptation of the pathogen. In *C. albicans,* glucose phosphorylation is mainly performed by the hexokinase 2 (*Ca*Hxk2). In addition, in the presence of glucose, *Ca*HxK2 migrates in the nucleus and contributes to the glucose repression signaling pathway. Based on the known dual function of the *Saccharomyces cerevisiae* hexokinase 2 (*Sc*Hxk2), we intended to explore the impact of both enzymatic and regulatory functions of *Ca*Hxk2 on virulence, using a site-directed mutagenesis approach. We show that the conserved aspartate residue at position 210, implicated in the interaction with glucose, is essential for enzymatic and glucose repression functions but also for filamentation and virulence in macrophages. Point mutations and deletion into the *N*-terminal region known to specifically affect glucose repression in *Sc*Hxk2 proved to be ineffective in *Ca*Hxk2. These results clearly show that enzymatic and regulatory functions of the hexokinase 2 cannot be unlinked in *C. albicans.*

## 1. Introduction

For microbial pathogens, the metabolic capacity to assimilate nutrients is a crucial factor to promote infection and commensal colonization. The opportunistic fungal pathogen *Candida albicans* colonizes diverse host microenvironments (skin, mucosa, blood, organs) [1] characterized by a highly variable carbon resource composition. To assimilate the spectrum of alternative carbon sources available and to support *C. albicans* proliferation in vivo, glyoxylate, lipid β-oxidation and gluconeogenic pathways are essential [2,3,4,5,6,7,8]. However, the ability to assimilate hexoses should not be neglected. Indeed, glucose, galactose and fructose are detected transiently in the gastrointestinal tract. Glucose (4–7 mM) is the only sugar present in blood [3]. Genomic tools revealed the expression of infection-associated genes involved in the glycolytic pathway during the colonization of kidneys and liver and for survival in blood [9,10,11]. During *Candida*-macrophage interactions, concurrent upregulation of glycolysis occurs in both host and pathogen, leading to glucose competition. So, by depleting glucose, *C. albicans* triggers rapid macrophage death [12].

In a previous study [13], we showed that the hexokinase *Ca*Hxk2 is essential for fitness and virulence in *C. albicans*, which was reconfirmed by the work of Wijnants et al. [14]. Glucose phosphorylation by *Ca*Hxk2 is not only necessary for glucose metabolism, but is also a key step for filamentation, stress response, macrophage and *Galleria melonnella* infection [13]. Moreover, *Ca*Hxk2 is implicated in regulatory functions. Upon growth in glucose (from 0.1% and more) *Ca*Hxk2 shuttles from the cytoplasm to the nucleus. In the absence of *Ca*Hxk2 (*Cahxk2∆/∆*), high affinity transporter gene expression is drastically enhanced after transfer on 2% glucose medium [13]. Therefore, besides its metabolic role as a glucose kinase, *Ca*Hxk2 also affects transcriptional regulation of glucose-repressible genes, suggesting a moonlighting function similar to that observed for *Sc*Hxk2 in *S. cerevisiae* [13].

In addition to its hexose kinase activity, the hexokinase *Sc*Hxk2 plays a regulatory role in glucose catabolite repression in *S. cerevisiae* [15,16,17,18,19,20]. The localization of *Sc*Hxk2 is glucose dependent. When grown in 2 to 4% glucose, about 15% *Sc*Hxk2 is localized in the nucleus [15,21]. The main role of *Sc*Hxk2 in glucose repression is reached by its interaction with the *Sc*Mig1 and *Sc*Snf1 proteins to constitute a repressor complex. Functional studies have revealed that, in the nucleus, *Sc*Hxk2 interacts with the *Sc*Mig1 transcriptional repressor that binds, with the *Sc*Snf1 protein kinase, to the promoter of most glucose-repressible genes [22,23,24]. Moreover, *Sc*Snf1 is a member of the AMPK-activated protein kinases, (AMPK) family, which includes highly conserved sensors of the cellular energy status in eukaryotes [25,26]. The interaction between *Sc*Hxk2, *Sc*Mig1 and the kinase *Sc*Snf1 has been particularly well described at the level of the *SUC2*-Mig1 repressor complex [27,28]. Under high glucose conditions, *Sc*Hxk2 stabilizes the repressor complex in the nucleus, preventing *Sc*Mig1 phosphorylation by *Sc*Snf1 kinase and thus maintaining its repression capacity [23,29]. Then, *Sc*Hxk2 adopts a close conformation that promotes its binding to *Sc*Mig1 and the assembly of the repressor complex [28]. When glucose is low or absent, *Sc*Snf1 becomes activated and phosphorylates *Sc*Mig1, which becomes then inactivated. Once phosphorylated by *Sc*Snf1, *Sc*Mig1 is exported to the cytosol, *Sc*Hxk2 does not enter the nucleus and repression is relieved. *Sc*Hxk2 translocation in the nucleus and its interaction with the transcription factor *Sc*Mig1 relies on a decapeptide located between K^7^ and M^16^ of the hexokinase and on the phosphorylation state of *Sc*Mig1 at S^311^ [27,29]. A *Schxk2* mutant, deleted of this decapeptide, revealed a loss of regulatory functions and a lack of interaction between *Sc*Hxk2 and *Sc*Mig1, while the deletion had no effect on the hexose kinase activity of the protein [27]. Moreover, the nuclear import of *Sc*Hxk2 requires the α/β-importin-dependent pathway and a lysin-rich nuclear localization sequence located between K^7^ and K^13^ [30]. Nucleocytoplasmic traffic of *Sc*Hxk2 is also regulated by phosphorylation and dephosphorylation of a serine residue at position 15 [31]. *Sc*Hxk2 is characterized by a monomer-dimer equilibrium in vivo; only the monomeric form is phosphorylated at the serine 15 position [21,32]. Relief from glucose repression is accompanied by the phosphorylation of the nuclear fraction of *Sc*Hxk2 at S^15^ by the *Sc*Tda1 protein kinase, translocation of the protein into the cytosol and loss of interaction with *Sc*Mig1 [33,34,35]. The dephosphorylation-mimicking mutant of *Sc*Hxk2 maintains its nuclear localization at low glucose concentrations, while a phosphomimetic mutant retains its cytoplasmic localization at high glucose levels [31]. Phosphorylation of S^15^ may favor the interaction of *Sc*Hxk2 with the nuclear receptor *Sc*Xpo1 and regulate the nuclear export of the protein [27].

In *S. cerevisiae*, the *Sc*Hxk2 regulatory and hexose-phosphorylating functions are decoupled and carried by different regions of the protein, defining it as a moonlight protein. Structural studies of *Sc*Hxk2 reveal a catalytic domain that binds ATP and hexose, allowing transfer of a phosphoryl group from ATP to the C-6 of the sugar [27,36]. Among the amino acids that constitute the glucose binding domain, four residues—S^158^, D^211^, E^269^, E^302^—interact directly with glucose in *Sc*Hxk2. Their mutation to A or G reduced the hexokinase activity by at least 90% [37].

In *C. albicans*, important players in glucose repression pathway are conserved. Among them, *Ca*Snf1 is a highly conserved regulator of nutrient stress responses and metabolic adaptation. Its activation by phosphorylation depends most likely on the key regulator *Ca*Sak1 [38]. Transcriptional profiling of *Casak1∆/∆* mutants showed that the kinase ensures basal expression of glyoxylate and gluconeogenesis genes even in glucose-rich media. Thus, *Ca*Sak1 contributes to the metabolic flexibility of *C. albicans* [38]. Regarding the downstream elements of the pathway, *Ca*Mig1 and *Ca*Mig2 are mediators of glucose repression in *C. albicans* [39,40,41]. Global analyses and comparisons of gene expression profiles in *Ca*Mig1 and *Ca*Mig2 mutants have revealed that both repressors have common but also selective targets, hexose transporters for *Ca*Mig1 and β-oxidation genes for *Ca*Mig2. However, *Ca*Mig1 and *Ca*Mig2 mostly function interchangeably, as shown by the little expression changes of upregulated genes in a double mutant *Camig1∆/∆Camig2∆/∆* strain or in either a *Camig1∆/∆* or a *Camig2∆/∆* single mutant [41]. Single and double mutants cause reduced damage in a macrophage model and show filamentation defects probably connected to the low expression of the major regulators of the filamentation process, *CaUME6* and *CaBGR1* [41].

However, data available in *C. albicans* do not focus on the role of the hexokinase *Ca*Hxk2 in the glucose repression pathway. Previously, we highlighted that glucose kinase and regulatory functions are conserved in *Ca*Hxk2 [13]. In the present study, our aim is to evaluate the impact of the regulatory functions of *Ca*Hxk2 on virulence of the pathogen. To dissociate enzymatic and regulatory functions of *Ca*Hxk2, we constructed site-directed mutants, on the basis of the extensive knowledge of the functional domains of hexokinase 2 in *S. cerevisiae* [23,27,30,31,37,42,43]. Our data reveal that the catalytic predicted domain of *S. cerevisiae* and *C. albicans* can be superimposed. However, mutations of the conserved serine at position 15 and deletion of the K^7^F^16^ decapeptide necessary for nuclear localization and interaction of the hexokinase with Mig1 in *S. cerevisiae* did not affect glucose repression, filamentation and virulence in *C. albicans*. Our data highlight significant differences between *Ca*Hxk2 and *Sc*Hxk2 regarding the role of the different identified domains of the protein implicated in glucose repression. Altogether, the moonlight protein model attributed to the *S. cerevisiae* hexokinase 2 is challenged in *C. albicans.*

## 2. Materials and Methods

### 2.1. Yeast Strains and Growth Conditions

*C. albicans* strains used in this study are listed in Appendix A. Yeast strains were maintained on YPG medium (1% yeast extract, 2% peptone, 2% glucose) at 30 °C. To analyze gene expression, wild type and mutant strains were grown on YPLac (glucose was replaced by 2% lactate) to early log phase (OD = 1.8) at 30 °C. Yeast cells were collected by centrifugation at 2500 rpm for 5 min, washed twice with sterile water and transferred in YPG (2% glucose) for 90 min. Cell samples were collected after 0, 45 and 90 min of growth on YPG. All cell samples were washed twice with sterile water and rapidly frozen at −80 °C. For filamentation experiments on solid medium, yeast strains were grown for 48 h on Spider medium (1% Nutrient Broth, 1% mannitol, 0.2% KH_2_PO_4_, 2% agar) and on YPNAG composed of 1% yeast extract, 2% peptone, 2.5 mM *N*-acetylglucosamine (GlcNAc) and 2% agar at 37 °C.

### 2.2. Construction of Mutant Strains

Mutant variants of the wild-type *CaHXK2* gene were made by site-directed mutagenesis using a QuickChange Site-Directed Mutagenesis Kit^®^ from Agilent Technologies (Les Ulis, France) according to the manufacturer’s instructions. Mutations were introduced in the pCaPC1c plasmid [13] (Appendix A) containing the wild type genomic region of *CaHXK2* spanning from –473 bp before the start codon to +363 bp after the stop codon and the *SAT1* flipper cassette from plasmid pSFS2A [44]. The plasmid was amplified with three couples of primers (Appendix A) to introduce the T^15^A, K^7^F^16^ and D^210^A mutations in the pCaHXK2T^15^A, pCaHXK2K^7^F^16^ and pCaHXK2D^210^A plasmids, respectively. Mutations were confirmed by sequencing each construct. Plasmids were digested by *Sph*1 and *Xma*1 to release the genomic regions containing the mutations to transform the *C. albicans Cahxk2∆/∆* mutant strain (Appendix A, Appendix A) [13]. *C. albicans* transformation was performed using the PEG Lithium technique [45]. After transformation, mixtures were incubated 2 days in YPG + nourseothricin 250 µg/mL (WernerBioAgent, Jena, Germany). Nourseothricin-sensitive cells were obtained according to Reuss et al. [44]. Transformants were grown overnight in YPG medium without selective pressure. Cells were plated on YPG containing nourseothricin (25 µg/mL). Small colonies containing nourseothricin-sensitive cells were selected after 2 days of growth at 30 °C. Both deleted *CaHXK2* alleles of the *Cahxk2∆/∆* mutant strain were complemented in a similar manner after elimination of the *SAT1* flipper cassette.

### 2.3. Preparation of Yeast Cell Extract

To prepare protein extracts, cells were suspended in 500 µL of 0.1 M Tris-HCL buffer supplemented with 10% phenylmethylsulfonylfluoride (PMSF) and then broken in the presence of 1.5 mL of glass beads using a FastPrep^®^-24 (MP Biomedicals; Illkirch, France) machine (five bursts 6.5 m/s for 30 s). Following this lysis step, cell extracts were centrifuged at 1500 rpm for 10 min at 4 °C. Proteins from the supernatant were quantified using NanoDrop 2000^®^.

### 2.4. Determination of Glucose Kinase Activity

The hexokinase activity was measured spectrophotometrically through NADP^+^ reduction in a glucose-6-phosphate dehydrogenase-coupled reaction. Each reaction was performed in a 1 mL spectrophotometer cuvette at room temperature. The final assay mixture consisted of 100 µL of 25 mM HEPES buffer pH 7.5, 100 µL of 10 mM MgCl_2_, 100 µL of 1 mM β-NAD, 500 µg of crude protein extract obtained from log phase cells grown in YPG (2% glucose), two units of glucose-6-phosphate dehydrogenase and 100 µL of 10 mM d-glucose. Reactions were started with the addition of 100 µL of 5 mM ATP. Absorbance was continuously recorded at 340 nm for 10–15 min. Glucose kinase activity was expressed as nmol of NADPH/min/µg of proteins. Activities were calculated from the mean of three independent experiments and expressed as a percentage of the activity measured with wild-type crude protein extract.

### 2.5. Determination of Glucose Concentration

To determine glucose consumption, YPG medium (0.5% glucose) was inoculated to OD = 0.2 with a stationary phase culture. Glucose concentration was measured every hour over a period of 10 h using the Glucose Colorimetric Assay Kit, GOD-POD method (Elabscience^®^, Houston, TX, USA). A total of 10 µL of culture supernatant was mixed with 500 µL of phenol and 500 µL of the enzymatic mix provided in the kit. The mix was vortexed and incubated at 37 °C for 25 min. The presence of glucose induces a pink colour. Glucose concentration was measured spectrophometrically at 505 nm and calculated according to the recommendations of the Glucose Colorimetric Assay Kit.

### 2.6. RNA Extraction and RT-qPCR Analysis

Total RNA was extracted from cells grown to OD_600_ ≈ 1.5 by the acid phenol method [46]. For reverse transcription-quantitative PCR (RT-qPCR) experiments, 10 μg of total RNA extract was treated with DNase I (Ambion, ThermoFisher Scientific, Courtaboeuf, France). Then, ReVertAid H Minus reverse transcriptase (ThermoFisher Scientific, Courtaboeuf, France) was used as described by the manufacturer to generate cDNAs. RT-qPCR experiments were performed with the CFX 96 Bio-Rad light cycler using SsoAdvanced Universal SYBR Green Supermix (Bio-Rad, Marnes-la-Coquette, France). Relative quantification was based on the 2Δ*CT* method using *CaACT1* (actin) as calibrator. The amplification reaction conditions were as follows: 95 °C for 1 min, 40 cycles of 95 °C for 15 s, 60 °C for 30 s, and the final step of 95 °C for 10 s. A melting curve was generated at 95° for 10 s and 65 °C for 5 s, with an increment of 0.5 °C until 95 °C at the end of each PCR cycle, to verify that a unique product was amplified. Primers used are presented in Appendix A.

### 2.7. Infection of Phagocytes with Yeasts

Macrophages from the J774A.1 (ATCC TIB-67) murine cell line were infected as previously described [47] in cRPMI medium (RPMI-1640 without phenol red and supplemented with 10% heat-inactivated fetal bovine serum, 1 mM sodium pyruvate and 2 g/L sodium bicarbonate) at 37 °C under 5% CO_2_. In brief, 2 × 10^5^ macrophages per well were adhered overnight in 96-well plates and infected with 1 × 10^6^ Calcofluor White (CFW)-labeled yeast cells in stationary phase in cRPMI medium supplemented with 5 mg/mL CFW. Interaction was followed over a 24 h time course experiment.

### 2.8. Flow Cytometry Analysis

Flow cytometry assays were conducted as previously described [47] using a FACSCantoII (Becton Dickinson, Le Pont de Claix, France). Macrophage viability and the ratio of macrophages engaged in phagocytosis were determined after 30 min, 4 h and 24 h of infection with CFW-labeled yeasts. Quintuplets of each condition were done for each experiment. After trypsin treatment, macrophages were labeled with 0.2 µg/mL anti-mouse CD16-APC (Biolegend, San Diego, CA, USA) (a membrane stain) and 0.2 µM calcein-AM (Sigma, ST Quentin Fallavier, France) (a marker of active metabolism). The percentage of macrophage viability was calculated using the number of macrophages positive for both fluorescences (anti-CD16-APC and calcein-AM) when infected with yeasts compared to the control uninfected macrophages. Phagocytosing macrophages were quantified as the percentage of the double-stained macrophages also positive for CFW fluorescence. Bilateral unpaired T-test was used to establish statistical significance with a significance level set at *p* < 0.05. Gating strategy was based on the specificity of each marker for a cell type, CFW stained yeast cells, calcein and anti-CD16-APC stained macrophages. The threshold for each fluorescence was determined based on positive and negative controls. Yeast without macrophages in the presence of CFW were used as positive control for CFW fluorescence. Macrophages alone in the presence of CFW were used as negative control. Cells in absence of CFW also served as negative controls. Yeast and hyphae were identified as positive for CFW fluorescence. Alive macrophages were first identified based on their double fluorescence for calcein and anti-CD16-APC.

## 3. Results

### 3.1. The Regulatory Domains and Catalytic Residues Are Conserved in CaHxk2

In *S. cerevisiae*, domains implicated in catalytic and regulatory functions were clearly identified [23,27,30,31,35,37,42,43]. Alignment of *S. cerevisiae* and *C. albicans* hexokinase 2 amino acid sequences revealed a high level of identity (70%) and showed that most of the identified domains are conserved in *C. albicans* (Figure 1). Consequently, we used the compiled knowledge about *Sc*Hxk2 to design a set of mutants in *C. albicans.*

Amino acids from K^7^ to M^16^ (-KKPQARKGSM-) are essential to the regulatory role of *Sc*Hxk2, in particular due to their interaction with the repressor *Sc*Mig1 [16,23,27]. The alignment (Figure 1) makes it possible to identify such a decapeptide (-QPAQKRKGTF-) at the same position in the *C. albicans* sequence. Moreover, the glucose-dependent nuclear import of *Sc*Hxk2 requires a lysine-rich nuclear localization sequence (NLS), located between K^7^ and K^13^ in the same decapeptide of the hexokinase sequence [30]. Such a sequence has also been identified in *Ca*Hxk2 between the residues P^8^ and F^16^ (-PAQKRKGTF-) [13]. Based on these findings, we constructed a site-directed mutagenesis deletion mutant (*Cahxk2∆K^7^F^16^*) lacking the decapeptide situated between K^7^ and F^16^ in *Ca*Hxk2.

In *S. cerevisiae*, the nucleoplasmic shuttling of *Sc*HxK2, induced by glucose levels, is regulated by phosphorylation and dephosphorylation at S^15^ [31,35]. *Sc*Hxk2 phosphorylation at S^15^ upon glucose limitation promotes export of the hexokinase by facilitating its association with XpoI, a member of the importin-β family [43]. Taking these data in account, the *Ca*Hxk2 T^15^ (Figure 1), which was the proposed phosphorylatable residue located at the same position as the S^15^ in *Sc*Hxk2 [13], was mutated by site-directed mutagenesis to a non-phosphorylatable alanine (*Cahxk2T^15^A*).

Other well-identified *Sc*Hxk2 residues are the residues predicted to interact with its substrate. The glucose binding domains include four residues thought to interact directly with glucose, (S^158^, D ^211^, E^269^, E^302^). Analysis of mutant *Sc*Hxk2 enzymes show that all four site-directed mutations reduced hexokinase activity by at least 90% [37]. The regions spanning these four residues are almost strictly identical in both sequences (Figure 1). In order to assess the impact of the phosphorylation function in glucose repression, we chose to substitute by site-directed mutagenesis D^210^ to an alanine in the *C. albicans* hexokinase sequence, to construct *Cahxk2D^210^A*.

### 3.2. The Aspartic Acid Residue D^210^ Is Implicated in Glucose Phosphorylation in C. albicans

The aspartic residue D^210^ is predicted to interact directly with glucose during the *S. cerevisiae* hexokinase reaction [36,37,48]. To verify that this conserved residue is also implicated in *Ca*Hxk2 catalytic activity, we measured the hexose kinase activity displayed by the mutant strain *Ca*hxk2D^210^A and compared it to the activity detected in the wild-type strain (SC5314), the double disrupted mutant *Cahxk2∆/∆,* and the site-directed mutagenesis-generated mutants *Cahxk2T^15^A* and *Cahxk2∆K^7^F^16^* (Figure 2a). Data obtained with the wild-type and deleted strains confirmed that *Ca*Hxk2 contributes to 65% of the glucose kinase activity [13]. Data obtained with *Cahxk2∆K^7^F^16^* and *Cahxk2T^15^A* mutants revealed that deletion of the K^7^F^16^
*N*-terminal decapeptide or the substitution of T^15^ to a non-phosphorylatable alanine had no impact on the catalytic activity of the enzyme. On the contrary, values obtained with *Cahxk2D^210^A* were statistically comparable to the data from de double-deleted strain *Cahxk2∆/∆*. This revealed that the conserved aspartic acid residue at position 210 is essential to *C. albicans* hexose kinase catalytic activity.

To confirm these results, we measured extracellular glucose consumption by the different mutants and wild-type strains over the course of time (Figure 2b). Cells previously grown in the presence of 2% lactate were then transferred in a 0.5% glucose medium. During the first 4 h, glucose consumption was very slow and only decreased by 18% over the same period for all strains. After 4 h of growth, a drastic decrease in glucose extracellular concentration was detected for the wild-type *Cahxk2∆*K^7^F^16^ and *Cahxk2T^15^A* strains, reaching 2.5% of the initial concentration after 6 h and 0% after 7 h. Glucose consumption by *Cahxk2D^210^A* and *Cahxk2∆/∆* was affected compared to the other strains. In these two mutants only 40% of the glucose was consumed after 6 h of growth and 100% was used after 8 h. These data confirm the crucial role of the aspartic residue D^210^ for the hexose kinase activity of *Ca*Hxk2 and consequently for glucose consumption.

### 3.3. The CaHxk2 Catalytic Residue D^210^ Is Involved in Glucose Repression in C. albicans But Neither the N-terminal Decapeptide K^7^F^16^ Nor the Threonine T^15^

In *S. cerevisiae*, the *N*-terminal decapeptide K^7^M^16^ and the serine residue at position 15 are essential to the regulatory functions of *Sc*Hxk2 [31,35]. To investigate the contribution of the *N*-terminal decapeptide K^7^F^16^ and the threonine residue T^15^ in glucose repression, we analyzed the expression of genes (*CaICL1* and *CaPCK1*) involved in alternative carbon source utilization, which are known to be expressed in *C. albicans* when glucose is not available [49,50,51]. The isocitrate lyase *Ca*Icl1 is the key enzyme of the glyoxylate cycle, a metabolic pathway that permits the use of two-carbon compounds as carbon source when they are the only sources of carbon and produces precursors of gluconeogenesis. *CaPCK1* encodes for the phosphoenolpyruvate carboxykinase, a gluconeogenic enzyme responsible for phosphoenolpyruvate synthesis, a key step for metabolic flexibility in *C. albicans* [3]. Moreover, the expression levels of *CaICL1* and *CaPCK1* are disturbed in a *Camig1mig2∆/∆* double mutant, which indicates that those two genes are good reporters for testing the glucose repression pathway efficiency in *C. albicans* [41].

Transfer of wild-type yeast cells from lactate to glucose drastically decreased the expression level of both genes to less than 10% (Figure 3). This low expression level was observed during the 90 min time of glucose exposure, indicating that under our experimental conditions, glucose repression occurred and was maintained. Surprisingly, identical expression profiles were observed for the two mutants *Cahxk2∆K^7^F^16^* and *Cahxk2T^15^A,* indicating that the deletion of the decapeptide or the substitution of the threonine residue at position 15 to an alanine had no impact on glucose repression of *CaICL1* and *CaPCK1* under our conditions. The impact of glucose exposure on *CaICL1* and *CaPCK1* expression in *Cahxk2D^210^A* led to extremely different profiles. Instead of being drastically reduced, the level of expression of both genes decreased by half after 45 min of exposure to glucose, to reach 85 to 90% of the level initially detected in 2% lactate after 90 min in the presence of glucose. Analysis of *CaICL1* and *CaPCK1* expression in the double-deletion mutant (*Cahxk2∆/∆*) revealed exactly the same profiles (Figure 3). These data suggest that a *C. albicans hxk2* mutant, affected for hexose phosphorylation, either because the hexokinase gene is absent or because its catalytic activity is strongly reduced, cannot properly set up and maintain glucose repression.

To better elucidate the impact of *Ca*Hxk2 point mutations on glucose repression efficiency, we checked the expression profile of the high affinity hexose transporter *CaHGT12* [52]. This transporter shows sequence similarities with glucose sensors in yeasts [53]. In a previous work, we showed that *Ca*Hxk2 had repressor functions on *CaHGT12* gene expression [13]. Among the *C. albicans* transporters, *CaHGT12* is a particular case in terms of its very high level of expression during the first early steps of phagocytosis by macrophages, therefore in a glucose-poor environment [7,13,54]. After culture of wild-type cells to exponential phase in 2% lactate, *CaHGT12* was expressed at a very low level, which was set as a reference (Figure 4). Unlike *CaICL1* and *CaPCK1*, *CaHGT12* is not expressed after this first culture, which is expected for a high affinity glucose transporter. After 45 min in the presence of 2% glucose, transcription of *CaHGT12* transiently increased (×50) to be repressed at 90 min (×11) as a result of glucose repression. Almost the same profiles were observed for *Cahxk2∆K^7^F^16^* and *Cahxk2T^15^A,* which suggests that these mutations had no impact on *CaHGT12* gene expression in the presence of glucose. In a *Cahxk2D^210^A* mutant, as in the hexokinase deletion mutant (*Cahxk2Δ/Δ*), *CaHGT12* transcript level in the presence of glucose is about twice as high as for the wild-type strain. Full glucose repression was not achieved after 90 min in the presence of 2% glucose. Transcription levels remain statistically equivalent after 45 and 90 min of growth in repressing conditions. These data suggest that, as for *CaICL1* and *CaPCK1,* glucose repression of *CaHGT12* requires a functional hexokinase function.

### 3.4. Filamentation Is Affected in Cahxk2D^210^A

Morphological switch is a determining factor of virulence in *C. albicans.* Hyphae are intrinsically invasive on solid media and express numerous virulence factors, such as adhesins, the toxin Ece1, or tissue degrading enzymes [55,56]. To explore the impact of glucose repression on virulence, we first checked the ability of the various *C. albicans* hexokinase mutants to undergo a yeast-to-hyphae transition. Hyphal formation was induced on two different media. Spider medium contains mannitol, which depends on the hexokinase step to be metabolized. YPNAG medium contains *N*-acetylglucosamine, which requires *Ca*Hxk1, but not *Ca*Hxk2, to be metabolized [13,57].

Similar to our previous observations [13], a *Cahxk2∆/∆* deletion mutant is not able to develop hyphae on Spider medium but can be filamentous if the inducing carbon source (YPNAG) does not require *Ca*Hxk2 to be metabolized (Figure 5). The link between the ability to phosphorylate glucose and filamentation was confirmed by the hypofilamentous behavior of *Cahxk2D^210^A*, which did not show any filaments at the periphery of the colony on Spider medium. After growth on YPNAG, *Cahxk2D^210^A* produced hyphae. This suggests that the lack of filamentation observed in that strain is most certainly linked to hexose phosphorylation defects. Indeed, filamentation was restored when the catalytic activity of the hexokinase was not necessary for catabolism. The mutant strains *Cahxk2∆K^7^F^16^* and *Cahxk2T^15^A* developed hyphae on Spider and YPNAG media (Figure 5). This suggests that the *N*-terminal decapeptide and the T^15^ residue of *Ca*hxk2 do not support any function in the filamentation process. Moreover, because there is also a glucose effect on *N*-acetylglucosamine catabolism in *C. albicans* [58,59], and because hexose kinase function and glucose repression are linked, it was not possible to evaluate separately the impact of the regulatory functions of *Ca*Hxk2 on filamentation.

### 3.5. Virulence in a Macrophage Model Is Affected in Cahxk2D^210^A Mutant

To explore the impact of *Ca*Hxk2 point mutations on virulence, we analyzed the ability of the mutants to kill macrophages at different interaction times using an in vitro model assay (Figure 6). First, our data showed that mutations did not modify macrophage association with yeasts for any strain (grey part of the bars in Figure 6). This suggests that none of the mutations performed have an impact on the recognition step. In a previous work, we showed that *CaHXK2* was under-expressed after 30 min of phagocytosis and suggested that this enzyme may not be necessary for the first step of infection [7].

After 4 h of phagocytosis, infections performed with *Cahxk2∆K^7^F^16^* and *Cahxk2T^15^A* did not reveal significant differences as compared to the wild-type strain. However, survival of macrophages was significantly higher when *Cahxk2D^210^A* or *Cahxk2∆/∆* were used. After 4 h of infection with *Cahxk2D^210^A*, the survival rate of macrophages was four times higher than that of the wild-type strain, while it was three times higher when the double-deletion mutant *Cahxk2Δ/Δ* was used. Our data clearly reveal that in our macrophage model, virulence of hexokinase mutants is linked to hexose kinase activity. *N*-terminal residues highlighted in *S. cerevisiae* (K^7^F^16^ decapeptide and T^15^) do not seem to be of primary importance in any function of the protein connected to virulence in macrophages.

Finally, after 24 h of infection, the survival rate was very low for all strains (4 to 6%). We can assume that at this step, most of the yeasts have escaped from the phagocyte [13,47]. Such a similar profile, observed for all strains, could indicate that the virulence of *Cahxk2D^210^A* or *Cahxk2Δ/Δ* mutants was only delayed. As previously suggested for *Cahxk2Δ/Δ* [13], the virulence defect associated with *Cahxk2D^210^A,* with reduced hexose phosphorylation capacities, could concern the fungal escape from macrophages rather than the initial phagocytosis step.

## 4. Discussion

A significant number of proteins can apparently assume more than one function and perform completely unrelated tasks. A large number of moonlighting proteins is found in the group of sugar metabolism enzymes, such as, for example, aldolase, enolase, pyruvate carboxylase or hexokinase [17,19,60,61]. In *S. cerevisiae*, hexokinase2 (*Sc*Hxk2) is a leading example of moonlighting proteins in yeasts. The use of a whole panel of techniques, like immunoblotting, enzymatic assays and fluorescence distribution, have shown unambiguously that in yeast cells growing in 2% glucose, about 15% of the Hxk2 protein pool is located in the nucleus [15,21]. Strategic domains of the protein, enabling nuclear import, export and interaction with the other partners of the glucose repression pathway, have been identified and extensively studied [23,27,28,31,35,43]. Moreover, the domains involved in regulatory functions of the hexokinase2 in *S. cerevisiae* do not match the key predicted domains for enzymatic activity interacting with glucose or ATP [37]. Thus, by generating sets of mutants encoding different protein variants, it is possible to map the regions that participate independently of the moonlight functions of *Sc*Hxk2 [18]. In a recent study, we showed that in *C. albicans, Ca*Hxk2 is functionally homologous to *Sc*Hxk2. *Ca*Hxk2 is necessary for glucose phosphorylation and is able to localize into the nucleus, suggesting that it might be involved in glucose repression [13]. Here, we further show by pairwise protein sequence alignment that the two independent domains required for hexose kinase activity and glucose repression in *Sc*Hxk2 are conserved in *Ca*Hxk2. Based on this knowledge, in order to evaluate their role in glucose repression and virulence, we wished to target separately the regulatory and enzymatic functions of the hexokinase 2 in *C. albicans* by constructing a set of point mutants.

In *S. cerevisiae,* the regulatory domain of *ScHxk2* involved in glucose repression spans from lysine 7 to methionine 16. *Sc*Hxk2 deleted from this region was shown to conserve full catalytic activity but could not translocate to the nucleus because the nuclear localization sequence (NLS) located in this specific region had been removed [30]. In the presence of glucose, *Sc*Hxk2 sequestration in the nucleus depends on its direct interaction with *Sc*Mig1 mediated by this regulatory domain [16,23,31] (Figure 7). Furthermore, in low glucose, *Sc*Hxk2 is phosphorylated in vivo within the regulatory region at S^15^, which enables the nuclear export of the protein and the disorganization of the *Sc*Mig1 repressor complex [31]. In *Ca*Hxk2, the regulatory region is conserved and spans from lysine 7 to phenylalanine 16, a region containing one of the two predicted NLSs of *Ca*Hxk2 (this study; [13]). This region contains a threonine at position 15 (similar to serine 15 of *Sc*Hxk2) that was shown to be phosphorylated in vivo in a phosphoproteome analysis of *C. albicans* [62]. Altogether, this suggests that, in *C. albicans*, the role of *Ca*Hxk2 in glucose repression might be achieved through a mechanism similar to that observed for *Sc*Hxk2. Consequently, the deletion of the regulatory/NLS region (*Cahxk2∆K^7^F^16^)* or the mutation of threonine 15 to a non-phosphorylatable alanine (*Cahxk2T^15^A)* should severely affect glucose repression but not hexose phosphorylation in *C. albicans*. Strikingly, while glucose phosphorylation was not affected in the *Cahxk2∆K^7^F^16^* and *Cahxk2T^15^A* mutants as observed in *S. cerevisiae,* glucose repression seems not to be affected by the mutation of the predicted regulatory region of *Ca*Hxk2. Indeed, the expression patterns of *CaPC*K1, *CaICL*1 and *CaHGT12*—three genes known to be repressed by glucose [13,41]—during a glucose shift were almost similar to the profiles observed in the wild-type strain. These data reveal that the model established in *S. cerevisiae* cannot be transposed to *C. albicans*, and the observed role of *Ca*Hxk2 in glucose repression should be achieved through a different molecular mechanism.

In *S. cerevisiae*, in high glucose concentrations, *Sc*Hxk2 adopts a closed conformation that promotes its binding to *Sc*Mig1, contributing to the formation of a transcriptional repressor complex [28]. Genetic and gene expression studies revealed that in *C. albicans*, *Ca*Mig1 and its paralog *Ca*Mig2 also mediate glucose repression. *Ca*Mig1 represses *CaHGT12,* while *CaICL1* and *CaPCK1* are repressed by *Ca*Mig2 and *Ca*Mig1 [41]. Our data strongly suggest that *Ca*Hxk2 contributes to the same glucose regulation of those three genes. However, in *C. albicans*, interactions between the hexokinase 2 and/or *Ca*Mig1 and *Ca*Mig2 have not been demonstrated. If an interaction takes place, the N-terminal decapeptide region of *Ca*Hxk2 should not be involved. Ultimately there may not be any interaction between *Ca*Hxk2 and *Ca*Mig1. Moreover, the existence of additional mediators of glucose repression could be considered. Verifying these hypotheses could explain that *S. cerevisiae* and *C. albicans* models cannot be superimposed. Further investigations regarding *Ca*Hxk2 interaction with potential contributing proteins, and in particular *Ca*Mig1 and *Ca*Mig2, will be necessary to address this question.

In *Cahxk2D^210^A*, site-directed mutation of one of the four residues predicted to interact with glucose in *Sc*Hxk2 [37], reduced glucose kinase activity to the same level as in a null mutant (*Cahxk2∆/∆*). First, this shows that D^210^ function is conserved in *S. cerevisiae* and *C. albicans* sequences and also that this residue is probably essential for glucose attachment and further phosphorylation in *C. albicans*. Moreover, contrary to *S. cerevisiae*, this data shows that glucose phosphorylation and glucose repression functions of the hexokinase 2 are linked in *C. albicans* and not allowed by separate domains. Data on structural requirements of *Sc*Hxk2 for triggering catabolite repression have revealed that, in most hexokinase mutants, there was no correlation between hexokinase activity in vitro, in vivo sugar phosphate accumulation and the establishment of catabolite repression [63,64,65,66]. This was also demonstrated in *Candida utilis* [67]. However, partially contradictory observations were proposed, and, nevertheless, a link between glucose phosphorylation and glucose repression was suggested. By using doxycycline-controlled systems of hexokinase expression and screening *S. cerevisiae* mutants through adaptative evolution under a mixture of xylose and the toxic glucose analog 2-deoxyglucose (which can be phosphorylated but not metabolized), the isolation of strains able to consume simultaneously glucose and xylose or galactose was achieved. This study revealed that the simultaneous uptake of glucose and other carbon sources (indicating glucose repression release) was directly connected to a reduced glucose metabolic flux. This was assigned to mutations in the glucose phosphorylating enzymes or a reduced hexokinase 2 gene expression level [68]. Furthermore, similarly to what we have shown for *Cahxk2D^210^A*, a recent study on 2-deoxyglucose resistance in *S. cerevisiae* identified a *Schxk2D^210^A* resistant mutant displaying a severe defect in both hexokinase activity and glucose repression [69]. These data suggest that the decoupling of enzymatic and catalytic functions of *Ca*Hxk2 may not be as clear-cut as proposed in *S. cerevisiae* and give credit to our findings.

Several hypotheses could explain why a phosphorylation defect impacts glucose repression. The AMP kinase *Sc*Snf1, a highly conserved energy sensor, is a central player in glucose repression. Upon glucose limitation, *Sc*Snf1 becomes activated and then phosphorylates *Sc*Mig1, leading to inhibition of glucose repression [20,24,70]. *Sc*Snf1 activation correlates with increased cellular AMP:ATP ratios, which are tightly regulated by glucose metabolism [70,71]. Glycolysis being one of the ways to produce ATP, a reduction of hexokinase activity could deplete ATP and increase the AMP:ATP ratio, leading to *Sc*Snf1 activation and glucose de-repression. In addition, by limiting the rate of glucose phosphorylation, the accumulation of intracellular glucose could increase, and the process of sensing through transporter preference could be altered. This could also affect the ability to modulate response to sugar ratios, possibly through the accumulation of endogenous hexokinase inhibitors [68].

In a previous paper, we showed that induction of filamentation requires upregulation of hexose kinase genes. Moreover, the double-deletion mutant *Cahxk2* retains the filamentation capacity when the carbon source does not require *Ca*Hxk2 to be assimilated [13]. Here we specify that glucose phosphorylation capacities of the hexokinase are crucial for filamentation. In the same way, glucose phosphorylation activity is crucial for macrophage killing. Morphological switch is a determining factor of virulence in a macrophage model because there is a strong correlation between hyphal growth and macrophage lysis [72]. In addition, glucose phosphorylation could impact hyphal development at the metabolic level. As described by Tucey et al. [12] in the context of macrophage infection, host and pathogen up-regulate glycolysis to compete for glucose. *C. albicans*-activated macrophages shift to Warburg metabolism and become dependent on glucose for survival. A glucose phosphorylation-deficient strain could not compete efficiently for glucose and might become hypo-virulent. Moreover, defects in both filamentation and virulence could be the consequence of an affected glycolytic flux in many cellular aspects. For example, osmolytes like trehalose, glycerol and arabitol depend on glucose-6-phosphate availability to be synthetized and stored in fungal cells subjected to oxidative, osmotic or thermal stresses that could be encountered during phagocytosis.

Finally, our findings show that glucose phosphorylation appears to be essential to glucose repression. The two functions being intimately linked, it was not possible for us to separately evaluate the role of the regulatory functions of *Ca*Hxk2 during infection. However, transcriptional response of *C. albicans* cells to macrophage phagocytosis has been extensively described as a drastic metabolic reprogramming event. When phagocytosed, *C. albicans* cells display a metabolic program whereby the glyoxylate cycle and gluconeogenesis are activated early, while the subsequent progression of systemic disease, tissue colonization, is dependent on glycolysis [2,3,7,8,12,73,74]. Glucose metabolism via glycolysis and fermentation appear to be crucial during gastrointestinal colonization [5,11]. In the more complex context of the mouse model of systemic candidiasis, the use of GFP fusions has provided direct evidence for a high degree of heterogeneity within fungal cell population. While, for example, *C. albicans* cells infecting mouse kidneys express mostly glycolytic genes, some cells switch to gluconeogenesis metabolism, depending on their immediate environment [75,76]. All these arguments show that during host infection *C. albicans* regulates central carbon metabolism in a niche-specific manner. The glucose repression pathway, to which *Ca*Hxk2 contributes, is obviously a central player in the adaptation process to contrasting host microenvironments encountered by the pathogenic yeast.

## Figures and Tables

**Figure 1 microorganisms-09-00848-f001:**
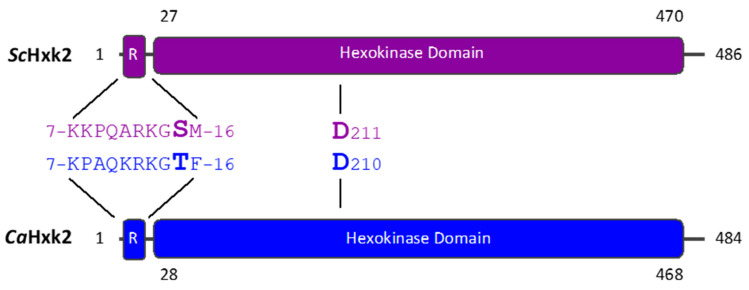
Schematic diagram showing the conserved residues in the *Sc*Hxk2 and *Ca*Hxk2 amino acid sequences and the position of the regulatory (R) and hexokinase domains. The position strictly conserved of D^211^ in *Sc*Hxk2 and D^210^ in *Ca*Hxk2 involved in glucose binding and situated in the hexokinase domain is indicated in bold. The *Sc*Hxk2 regulatory region (K^7^M^16^) is conserved in *Ca*Hxk2 (K^7^F^16^). The phosphorylated S^15^ known to control *Sc*Hxk2 localization is aligned with T^15^ in *Ca*Hxk2 (in bold). In the *Cahxk2∆K^7^F^16^* mutant, the whole decapeptide situated between the lysine residue at position 7 and phenylalanine at position 16 was deleted. For strains *Cahxk2T^15^A* and *Cahxk2D^210^A*, the threonine residue at position 15 and the aspartic acid at position 210 have been replaced by alanine residues. Numbers in black close to the boxes indicate their position within the amino acid sequence. Alignment was performed with MBOSS Needle.

**Figure 2 microorganisms-09-00848-f002:**
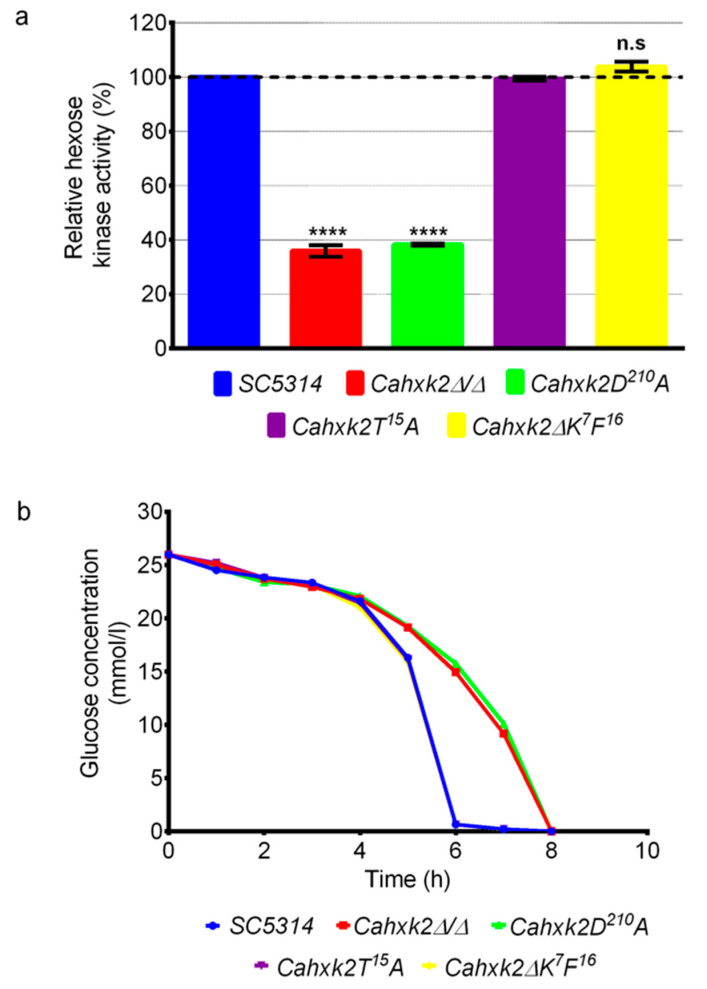
Relative hexose kinase activity and glucose consumption by *C. albicans* wild-type strain (SC5314), *Cahxk2* deletion (*Cahxk2Δ/Δ*) and punctual mutants (*Cahxk2∆K^6^F^16^*, *Cahxk2T^15^A* and *Cahxk2D^210^A*). (**a**) Hexose phosphorylation activity in wild-type and mutant *C. albicans* cell extracts obtained from YPG log phase cultures (2% glucose). For each strain, the amount of glucose-6-phosphate produced was measured and expressed as a percentage of the wild-type strain. Data are presented as a mean (+standard deviation) of three independent experiments performed on three different biological samples, in triplicates (*n* = 9). **** *p* < 0.0001; one-way ANOVA using Tukey’s method. n.s. = not significant. (**b**) Glucose consumption over a period of eight hours in 0.5% glucose medium. YPG (0.5% glucose) was inoculated to OD = 0.2 with stationary phase cells grown in 2% lactate. Glucose content, before inoculation, was measured to 25.96 mm/L. Data represent the mean of three experiments.

**Figure 3 microorganisms-09-00848-f003:**
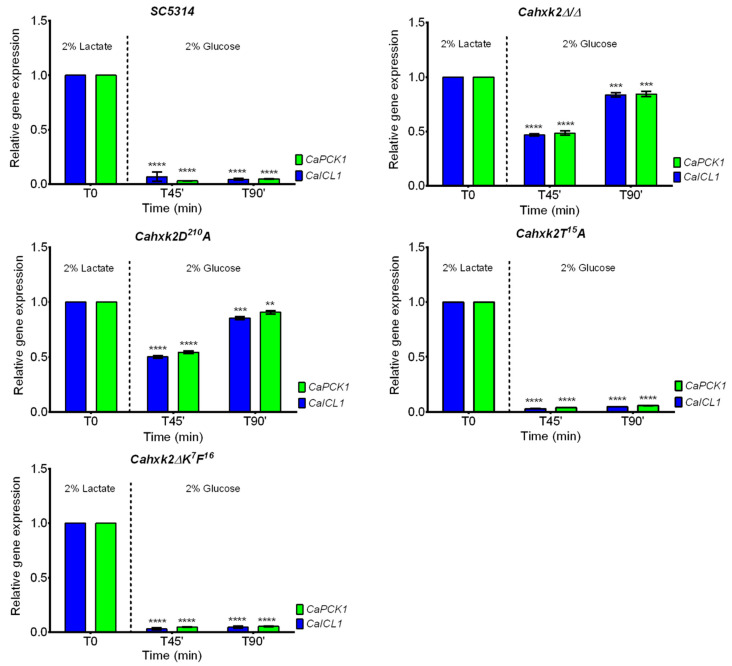
Relative gene expression of *CaICL1* and *CaPCK1* in wild-type (SC5314) and *CaHXK2* deletion (*Cahxk2Δ/Δ*) and punctual mutants (*Cahxk2∆K^6^F^16^*, *Cahxk2T^15^* and *Cahxk2D^210^A*) during glucose repression. Cells were grown in the presence of 2% lactate to OD = 1.2. Glucose 20% was then added to reach the final concentration of 2%. Cells were collected after 45 and 90 min. Transfer from a carbon source to another is indicated by a dotted line. The level of *CaICL1* and *CaPCK1* mRNAs was expressed relative to their abundance in 2% lactate (T0), which was set to 1. The expression was normalized to the level of the *CaACT1* mRNA internal control. Results represent a mean (+ standard deviation) of three independent experiments performed on three different biological samples, in duplicate (*n* = 6). **** *p* < 0.0001, *** *p* = 0.0002, ** *p* = 0.002. *p*-Values were calculated by one-way ANOVA using Tukey’s method.

**Figure 4 microorganisms-09-00848-f004:**
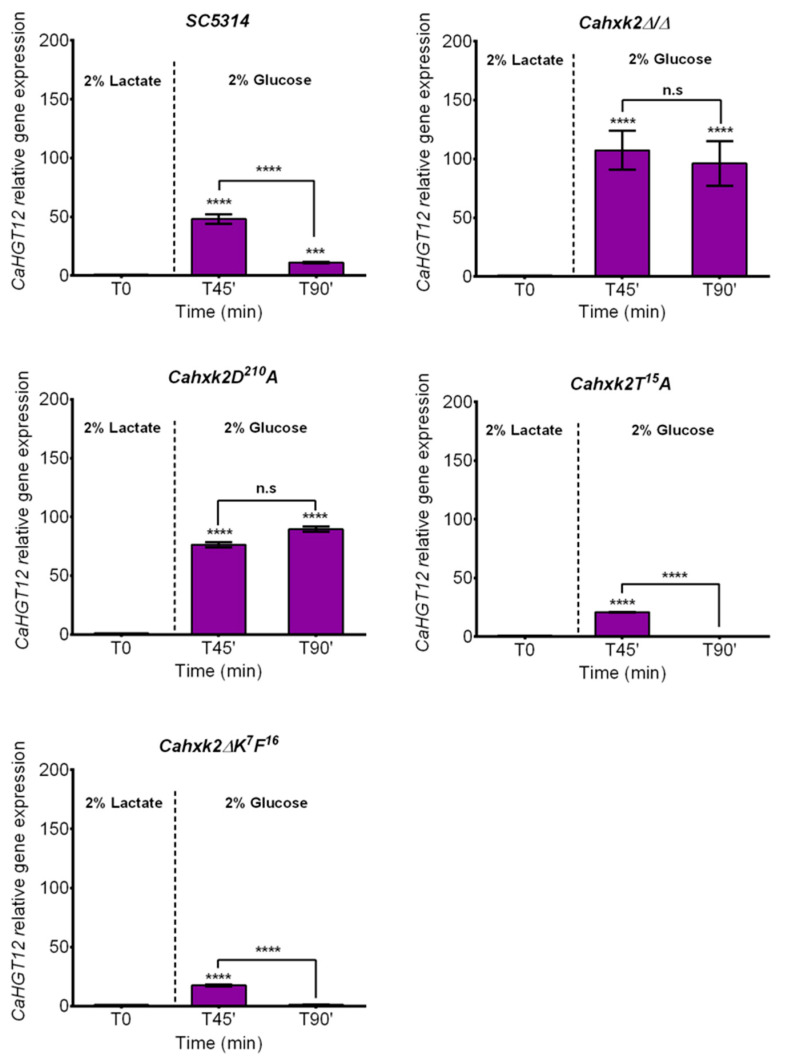
Relative gene expression of *CaHGT12* in wild-type (SC5314) and *Cahxk2* deletion and punctual mutants (*Cahxk2Δ/Δ*, *Cahxk2∆K^6^F^16^*, *Cahxk2T^15^A* and *Cahxk2D^210^A*) during glucose repression. Cells were grown in the presence of 2% lactate to OD =1.2. Glucose 20% was then added to reach the final concentration of 2%. Cell samples were collected after 45 and 90 min. Transfer from a carbon source to another is indicated by a dotted line. The level of *CaHGT12* mRNAs was expressed relative to their abundance in 2% lactate, which was set to 1 (T0). The expression was normalized to the level of the *CaACT1* mRNA internal control. Results represent a mean (+ standard deviation) of three independent experiments performed on three different biological samples, in duplicate (*n* = 6). N.s.: non-significant, **** *p* < 0.0001, *** *p* = 0.0002. *p*-Values were calculated by one-way ANOVA using Tukey’s method.

**Figure 5 microorganisms-09-00848-f005:**
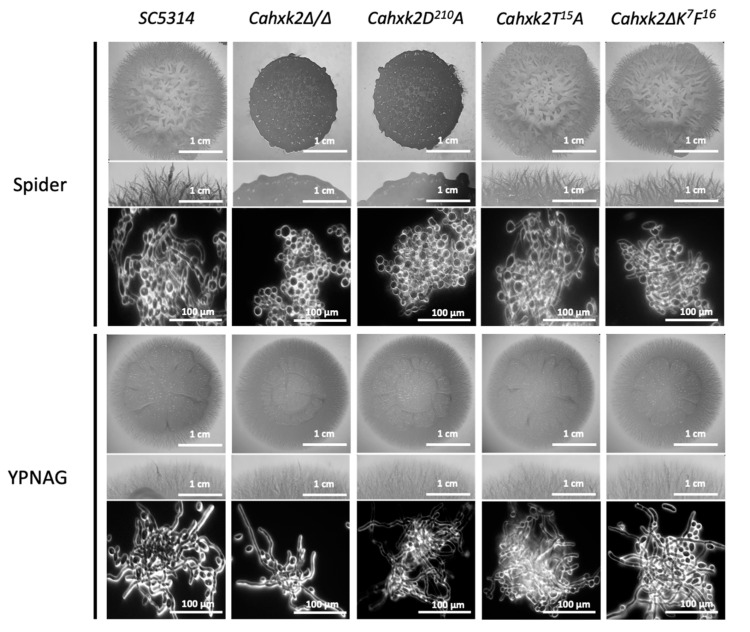
Filamentation of *Cahxk2* mutants. *C. albicans* wild-type and mutant strains were grown 3 days at 37 °C on Spider and YPNAG media. For each medium the upper and middle panels show photographs of the macroscopic appearance of the colonies. The lower panels were obtained using Zeiss Askioscop 2 Plus microscope and show the microscopic aspect. Filamentation assays were repeated three times.

**Figure 6 microorganisms-09-00848-f006:**
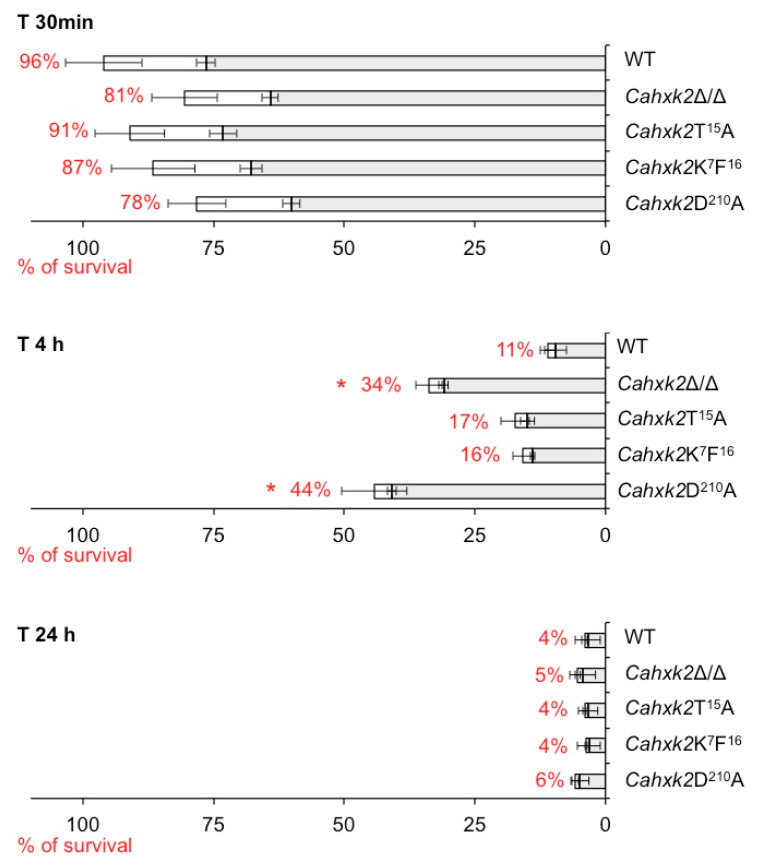
*Cahxk2D^210^A* mutant is hypo-virulent in a macrophage model. Flow cytometer analysis of mouse macrophage interaction with live *C. albicans* cells in stationary phase at MOI 1:5 (1 macrophage for five yeasts) over a 24 h time course experiment. The horizontal bars represent the macrophage survival, indicated as a percentage on the left side of the bar. The white part of the bars represents the percentage of non-phagocytosing macrophages. The shaded part represents the percentage of phagocytosing macrophages. Histograms represent a mean of three independent experiments performed on two different biological samples, repeated five times (+ standard error). Unpaired T-test was used to establish statistical significance. * *p* < 0.005.

**Figure 7 microorganisms-09-00848-f007:**
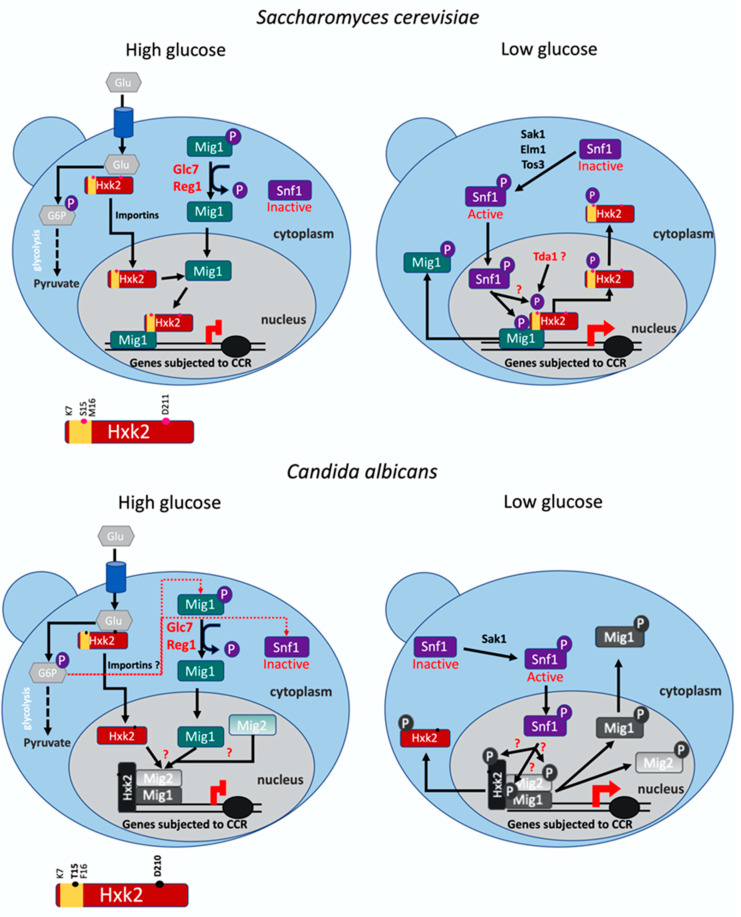
Schematic representation of the main elements implicated in the hexokinase2 (Hxk2) moonlighting functions in *C. albicans* and *S. cerevisiae.* Hxk2 is represented by a red rectangle. Point mutations realized in the *C. albicans* sequence (this study) are indicated by black spots (D^210^, T^15^) and by red spots in the *S. cerevisiae* sequence. The *N*-terminal deleted region from K^7^ to F^16^ in *C. albicans* (this study) and from K^7^ to M^16^ in *S. cerevisiae* is delimited in yellow. According to the *S. cerevisiae* model, under high glucose conditions, Hxk2 shuttles from cytoplasm to the nucleus via importins, where it stabilizes the repressor complex, preventing Mig1 phosphorylation by Snf1. The *N*-terminal decapeptide K^7^ to M^16^ is necessary for nuclear import and interaction with Mig1. Therefore, expression of genes subjected to carbon catabolite repression (CCR) is repressed. When glucose is low, Snf1 becomes activated and phosphorylates Mig1, which becomes inactivated and leaves the nucleus. Hxk2 is phosphorylated at S^15^ and exported to the cytosol. Targeted genes are then expressed. In *C. albicans*, under high glucose condtions, Hxk2 joins the nucleus. *C. albicans* D^210^, implicated in glucose binding and phosphorylation, is necessary for carbon catabolite repression. The *N*-terminal decapeptide K^7^ M^16^ and residue S^15^ are not involved glucose repression in *C. albicans*. The interaction between Hxk2, Mig1 and/or Mig2 has not yet been demonstrated, but their respective contribution to glucose repression has been shown. At low glucose conditions, Snf1 is activated by Sak1. The nuclear localization of Snf1 and its interaction with the repressor complex have not yet been shown. Localization of elements that are indicated in grey or black has not yet been demonstrated. Interactions that are deduced from the model *S. cerevisiae* but have not been shown in *C. albicans* are indicated by red question marks. Red dotted lines indicate the coupling between enzymatic and regulatory functions.

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
