# Peer review of "Candida albicans Hexokinase 2 Challenges the Saccharomyces cerevisiae Moonlight Protein Model"

_microorganisms, 2021, doi:10.3390/microorganisms9040848_

Round 1
Reviewer 1 Report
In the manuscript “Candida albicans hexokinase 2 challenges the Saccharomyces cerevisiae moonlight protein model” authors analyze very interesting phenomenon of protein moonlighting in relation to the enzyme from basic cellular metabolism and they indicate differences between fungal species in the regulation of additional functions performed by one protein.
The manuscript is well written and clearly arranged. I have no major concerns, but I could propose to present an additional figure in the discussion section, where authors could schematically compare the molecular basis of regulation of the enzymatic and regulatory functions of Hxk2 in C. albicans and S. cerevisiae.
Author Response
Response to reviewer1.
Dear reviewer,
Thank you for your answer and helpful comments concerning our manuscript.
“I could propose to present an additional figure in the discussion section, where authors could schematically compare the molecular basis of regulation of the enzymatic and regulatory functions of Hxk2 in C. albicans and S. cerevisiae.”
As proposed by reviewer 1, we added a schematic representation of the main elements implicated in the hexokinase2 moonlighting functions of in C. albicans and S. cerevisiae.
A figure (figure 7) was added in the discussion section. It was mainly dedicated to the elements cited in the manuscript. We had to mix two models, two glucose conditions, data at different levels, like cellular localization, signalisation and catalytic functions, details concerning also amino acid residues, which makes the scheme dense. Then, we propose this new figure that we hope will bring a more synthetic view to the discussion .
With regards
Pascale Cotton

Reviewer 2 Report
In this manuscript, Laurian et al. identified conserved regions in C. albicans involved in either the enzymatic and regulatory activity of the ScHxk2. To analyze whether these regions are also having a role in the dual functions of CaHxk2, they created site-directed mutations. While the K7F16 N-terminal decapeptide and T15 seem not to be involved in glucose repression in C. albicans, the D210 not only showed hexokinase catalytic activity but also a role in glucose repression. These results provide enough evidence to conclude that the model established in S. cerevisiae cannot be applied in C. albicans. Additionally, the authors show that D210 is also involved in the ability of the fungus to produce filaments or kill macrophages in conditions where the catalytic activity of Hxk2 is required.
In general, the antecedents and goals of this manuscript are well presented and are clear. The conclusions are reasonable. However, a survival analysis using a murine model would have strengthened the role of D210 in virulence. Is Cahxk2∆/∆ or Cahxk2D88A avirulent in a murine model?
Finally, I wonder if the defects on both filamentation and virulence are linked to the trehalose biosynthetic pathway as G6P is the substrate for this pathway involved in stress resistance, filamentation or virulence.
Minor comments
- Throughout the ms, authors use both 1 and 3 letter code notations for aa. Please, use only one code notation.
- Line 327. “…or the substitution of Ser15 to a non phosphorylatable Alanine had no impact on the…”. It is T15.
- Line 362. CaIcl1 or CaICL1?
- Line 585. “…Yeast cells, when grown on glucose uses glycolysis as primary method of ATP production.” Revise.
Author Response
Response to reviewer 2
Dear reviewer,
Thank you for your answer concerning our manuscript. The suggestions offered have been very helpful. We have taken into consideration all the comments and aimed to answer.
Due to formatting problems especially concerning the italicized characters and the name of the strains, the corrected version contains those corrections in addition to the corrections asked by the reviewers.
-Nevertheless, the only mutant affected in the catalytic activity that the authors present is the D210A, which they report being a peculiar mutant in S. cerevisiae as well since it affects both Hxk2 in this yeast as well. This is not sufficient to discriminate if the catalytic activity of CaHxk2 is required for glucose repression in Candida. At least another mutant affected in catalytic activity that in S. cerevisiae has no defects in glucose repression should be tested, to definitely prove that the catalytic activity of CaHxk2 is required for glucose repression as well.
We focused our work on the deletion of the N-terminal decapeptide (from K7 to M16) because this deletion has been tested and analyzed in several S. cerevisiae studies (references 27, 29, 30). It has been confirmed that this deletion impacts both the interaction with ScMig1 and the sequestration in the nucleus. Because the N-terminal decapeptide region of CaHxk2 (from K7 to F16) presents a strong homology with the S. cerevisiae N-terminal decapeptide sequence it seemed appropriate to us to construct such a mutant. The same comment could be done concerning Cahxk2T15A.The CaHxk2 T15 phosphorylatable residue located at the same position than the S15 in ScHxk2 was mutated to a non-phosphorylatable alanine (Cahxk2T15A). In S. cerevisiae this mutant has been characterized on several occasions (references 27, 31, 32, 35). Here again it seemed appropriate to us to construct such a mutant.
In S. cerevisiae, as suggested by reviewer 2, a Hxk2wca mutant (Pelaez et al., 2010) for which the last eight amino acids have been deleted and also the S304 residue has been replaced by a phenylalanine, has also been characterized. Analysis confirmed that the two mutations play a role in the catalytic activity but not in the regulatory functions of ScHxk2. However, because the S304 residue is not conserved in the C. albicans sequence (an isoleucine is present instead of a serine) we did not choose to construct that mutant.
Concerning the Cahxk2D210A mutant, its phosphorylation capacities were statistically comparable to the data from the double deleted strain (Cahxk2∆/∆). Consequently, we felt this mutant was sufficient to show the link between glucose kinase activity and glucose repression.
I would also add that the experiments presented in our article were part of a funded program and grant that ended now. It will not be possible to us to construct another mutant within an acceptable period of time.
Minor remarks
-Fig. 1 is too large and the legend is difficult to read.
Figure 1 has been changed for a more graphic and synthetic one. Legend has been modified according to it.
-Line 68. ‘Get’ should be ‘gets’. Change has been made, line 69.
-Line 196. ‘specific’ should be ‘unique’. Change has been made. Now line 216.
-Lines 319-331. There was a formatting problem, I think. This is not the entire subtitle, is it?
Yes indeed, there was a formatting problem, not only the title but the whole paragraph was in italics. Correction has been made. Now lines 385 to 437.
Line 358-359. ‘To investigate the contribution of the N-terminal decapeptide K7F16 and the serine residue Ser15 in glucose repression, we analyzed…’. Since the authors are investigating CaHXK2 protein, they should not refer to amino acids in the same position on ScHxk2. It is confusing. Ser15 should be T15. The same in the title of the sub-paragraph.
I fully agree, changes have been made T15 instead of S15 in the title (line 466) and in the paragraph (line 469).
With regards
Pascale Cotton

Reviewer 3 Report
The manuscript entitled ‘Candida albicans hexokinase 2 challenges the Saccharomyces cerevisiae moonlight protein model’ by Romain LAURIAN , Jade RAVENT , Karine DEMENTHON , Marc LEMAIRE , Alexandre SOULARD , Pascale COTTON gives an interesting novel view on the activity of the moonlight protein Hexokinase 2, which has been well characterized in budding yeast but not in Candida. In fact, the authors demonstrate that the separation of function and the working model defined in budding yeast and based on a nuclear role for Hxk2 in glucose repression pathway, distinct from the catalytic activity of the protein, is not valid for Candida albicans.
The manuscript is well written and the presented data are convincing in demonstrating that the motif that is involved in the regulation of nuclear import/export and Mig1 interaction in S. cerevisiae has no role in glucose repression in Candida albicans. Nevertheless, the only mutant affected in the catalytic activity that the authors present is the D210A, which they report being a peculiar mutant in S. cerevisiae as well since it affects both Hxk2 in this yeast as well. This is not sufficient to discriminate if the catalytic activity of CaHxk2 is required for glucose repression in Candida. At least another mutant affected in catalytic activity that in S. cerevisiae has no defects in glucose repression should be tested, to definitely prove that the catalytic activity of CaHxk2 is required for glucose repression as well.
Minor remarks
Fig. 1 is too large and the legend is difficult to read. Better add a graphic legend for the box marks.
Line 68. ‘Get’ should be ‘gets’
Line 196. ‘specific’ should be ‘unique’
Lines 319-331. There was a formatting problem, I think. This is not the entire subtitle, is it?
Line 358-359. ‘To investigate the contribution of the N-terminal decapeptide K7F16 and the serine residue Ser15 in glucose repression, we analyzed…’. Since the authors are investigating CaHXK2 protein, they should not refer to amino acids in the same position on ScHxk2. It is confusing. Ser15 should be T15. The same in the title of the sub-paragraph.
Author Response
Response to reviewer 3
Dear reviewer,
Thank you for your answer concerning our manuscript. The suggestions offered have been very helpful. We have taken into consideration all the comments and aimed to answer.
Due to formatting problems especially concerning the italicized characters and the name of the strains, the corrected version contains those corrections in addition to the corrections asked by the reviewers.
-However, a survival analysis using a murine model would have strengthened the role of D210 in virulence. Is Cahxk2∆/∆ or Cahxk2D210A avirulent in a murine model?
Of course, it will be interesting to know if a mutant affected in glucose repression and phosphorylation is clearly affected in a complex model like the murine model. On the other hand, in the current stage of progress of our work, the impact of glucose repression only, cannot be evaluated.
We do not have the opportunity to test our mutants in a murine model in the context of our laboratory. Moreover, this would have led to complex administrative and ethical procedures. We have chosen the macrophage model because we have some expertise with this model (Dementhon et al., 2012). We have also realized previous studies in the context of the hexokinase mutant using this model, leading to very reliable and clear results. Relevant works in the field of C. albicans glucose (and carbon sources) metabolism in the context of macrophage phagocytosis are available (among them: Tucey et al., 2018, Barelle et al., 2006; 2004; Lorenz and Fink 2002) which allowed us to refer to these studies to discuss our data. Moreover, because the macrophage model is a single cell system, C. albicans can only be inside the cell or outside, in the infection medium. Testing the virulence of our strains in macrophages appeared to us simpler than in a complex organism, with contrasting microenvironments in terms of the carbon source.
-Finally, I wonder if the defects on both filamentation and virulence are linked to the trehalose biosynthetic pathway as G6P is the substrate for this pathway involved in stress resistance, filamentation or virulence.
Indeed, a lack of thehalose, or other sugars directly connected to glycolysis like glycerol, mannitol, arabitol, would affect survival under stress conditions like thermal, oxidative and oxidative stresses. However, we do not have here specific informations on the impact of hexokinase mutations on their synthesis and amount in mutant cells. In a previous paper (Laurian et al., 2019) we have shown that glucose phosphorylation defects affect stress resistance, filamentation and virulence in C. albicans. We discussed in that paper the fact that biosynthesis of osmolyte sugars and polyols, directly connected to the upper part of the glycolytic pathway, depends on glucose-6-phosphate availability, leading then, an hexokinase mutant to be hypovirulent. Moreover, carbon source could modulate cell wall architecture. Glucan synthases assemble UDP glucose residues produced from phosphorylated glucose. Absence of glucose repression could also redirect carbon flux. Glucose kinase enzymes exert certainly broader functions than phosphorylation and glucose repression.
I fully agree that It could be necessary to complete the arguments concerning filamentation and virulence defects in connection to hexokinase mutation.
We added the following sentence in the discussion section:
“Moreover, defects in both filamentation and virulence could be the consequence of an affected glucolytic flux on many cellular aspects. For example, osmolytes like trehalose, glycerol and arabitol, depend on glucose-6-phosphate availability to be synthetized and stored in fungal cells subjected to oxidative, osmotic or thermal stresses that could be encountered during phagocytosis”. (lines 799-803)
Minor comments
-Authors use both 1 and 3 letter code notations for aa. Please, use only one code notation.
Changes have been made throughout the manuscript for one letter code notation.
-Line 327. “…or the substitution of Ser15 to a non phosphorylatable Alanine had no impact on the…”. It is T15. I agree, change has been made, now line 433.
-Line 362. CaIcl1 or CaICL1? It is CaIcl1, the protein is concerns (line 472)
- Line 585. “…Yeast cells, when grown on glucose uses glycolysis as primary method of ATP production.” Revise. The sentence was changed for : « Glycolysis being one of the way to produce ATP, a reduction of hexokinase activity could deplete….. » lines 794.
With regards
Pascale Cotton

Round 2
Reviewer 3 Report
Thank you for your response. The paper has been improved by further comparison with the S. cerevisiae model.
Still some typing errors to be checked.
Best regards